# Acute Kidney Injury and Chronic Kidney Disease and Their Impacts on Prognosis among Patients with Severe COVID-19 Pneumonia: An Expert Center Case–Cohort Study

**DOI:** 10.3390/jcm13051486

**Published:** 2024-03-05

**Authors:** Jakub Klimkiewicz, Anna Grzywacz, Andrzej Michałowski, Mateusz Gutowski, Kamil Paryż, Ewelina Jędrych, Arkadiusz Lubas

**Affiliations:** 1Department of Anesthesiology and Intensive Care, COVID-19 Hospital, Military Institute of Medicine-National Research Institute, Szaserów 128 Str., 04-141 Warsaw, Poland; amichalowski@wim.mil.pl (A.M.); mgutowski@wim.mil.pl (M.G.); kparyz@wim.mil.pl (K.P.); 2Department of Nephrology, Internal Diseases and Dialysis, COVID-19 Hospital, Military Institute of Medicine-National Research Institute, Szaserów 128 Str., 04-141 Warsaw, Poland; agrzywacz@wim.mil.pl (A.G.); ejedrych@wim.mil.pl (E.J.); alubas@wim.mil.pl (A.L.)

**Keywords:** acute kidney injury, COVID-19, pneumonia, mortality, risk factors

## Abstract

**Background**: Acute kidney injury (AKI) is associated with substantial mortality. In this case–control study, we analyzed the impacts of AKI and chronic kidney disease (CKD) on outcomes in a group of 323 patients with severe COVID-19. The correlation of clinical and laboratory data with AKI and CKD was also analyzed. **Methods:** A retrospective case–control study was conducted among AKI, CKD, and normal kidney function (NKF) groups hospitalized in a COVID-19 center in 2021. **Results:** AKI patients had higher in-hospital mortality (55.2 vs. 18.8%, *p* < 0.001), more frequent transfers from the HDU to ICU (57.5 vs. 12.9%, *p* < 0.001), and prolonged hospital stays (15.4 ± 10.7 vs. 10.7 ± 6.7 days, *p* < 0.001) compared to the NKF group. AKI was a predictor of death (OR 4.794, 95%CI: 2.906–7.906, *p* < 0.001). AKI patients also had broader lung parenchymal involvement and higher inflammatory markers compared to the NKF group. Patients with prior CKD had higher in-hospital mortality compared to the NKF group (64.0 vs. 18.8%, *p* < 0.001, OR 4.044, 95%CI: 1.723–9.490, *p* = 0.013); however, transfers from the HDU to ICU were not more frequent (16.0 vs. 12.9%, *p* = 0.753). **Conclusions**: AKI among COVID-19 patients was correlated with more ICU transfers, higher morbidity, and greater markers of severe disease. Patients with CKD had a higher mortality; however, the rate of ICU transfer was not substantially higher due to their poor prognosis.

## 1. Introduction

Acute kidney injury (AKI) is a clinical syndrome characterized by a rapid deterioration in kidney function with elevations in serum creatinine and/or decreased urine output [1]. The term “acute kidney injury” replaced the previously used term “acute kidney failure” in 2004, which currently corresponds to stage 3 AKI.

The first AKI classification was named after the acronym RIFLE (Risk of renal impairment, Injury of the kidney, Failure of renal function, Loss of kidney function, and End-stage renal disease). It is divided into three levels of renal dysfunction: risk (increased serum creatinine 1.5 × baseline, a GFR decrease > 25%, or a urine output < 0.5 mL/kg/h for 6 h), injury (increased serum creatinine 2 × baseline, a GFR decrease > 50%, or a urine output < 0.5 mL/kg/h for 12 h), and failure (increased serum creatinine 3 × baseline, a GFR decrease > 75%, a serum creatinine ≥ 4.0 mg/dL with an acute increase ≥ 0.5 mg/dL, a urine output < 0.3 mL/kg/h for 24 h, or anuria for 12 h), and two clinical outcomes: loss (complete loss of kidney function for over 4 weeks) and end-stage renal disease (the need for renal replacement therapy (RRT) for > 3 months) [2]. These criteria are too complicated and require long follow-up treatment of the patients to assess long-term outcomes; therefore, they were not widely adopted. To facilitate and improve patient care, in 2007, the Acute Kidney Injury Network (AKIN) proposed a new AKI classification for adults based on the RIFLE criteria [3]. It included serum creatinine increases and reductions in urine output but not GFR evaluations. Many studies do not use urinary output measurements, mainly due to problems with the reliability of monitoring hourly diuresis [4,5,6,7]. Moreover, during the first evaluation, the previous serum creatinine level is often unknown. Serum creatinine may also be persistently elevated in people with chronic kidney disease (CKD) [4]. Currently, the 2012 KDIGO criteria are commonly used for the diagnosis and staging of AKI [4]. The risk of AKI increases with the severity of the patient’s condition. The occurrence of AKI is a major risk factor for death among patients admitted to the hospital for various reasons [2,3,4]. Due to different strains of SARS-CoV-2 being dominant in different geographical locations, varied organization of healthcare, and variations in the criteria used for hospitalization and ICU admission, the data concerning morbidity and mortality among COVID-19 patients show substantial differences. Taking the above information into consideration, we decided to perform a retrospective case–control study to investigate the clinical significance of AKI, as well as previous CKD, among hospitalized patients with severe COVID-19. The secondary goal was to identify factors associated with AKI among COVID-19 patients.

## 2. Materials and Methods

### 2.1. Patients

This study included Caucasian patients with COVID-19-related pneumonia who were hospitalized in Warsaw, Poland, during the SARS-CoV-2 variant delta pandemic outbreak from March 2021 to June 2021. We only enrolled patients in this study with “severe COVID-19 pneumonia,” defined as lung involvement and a need for any form of oxygen therapy (simple mask, non-rebreather masks, high-flow nasal oxygen therapy, or non-invasive or invasive mechanical ventilation). The exclusion criteria consisted of a lack of adequate kidney function monitoring during the hospital stay, previous RRT, and previous use of immunosuppressive therapies. Comorbidities were identified through the review of the patient’s electronic medical records.

All patients admitted to the COVID hospital, comprising 48 high-dependency unit (HDU) and 12 intensive care unit (ICU) beds, had a SARS-CoV-2 infection confirmed with real-time polymerase chain reaction (RT-PCR). All patients were managed according to up-to-date guidelines. Patients requiring simple oxygen masks, non-rebreather masks, or high-flow nasal oxygen therapy stayed in the HDU. In contrast, patients with profound hypoxemia requiring non-invasive mechanical ventilation, tracheal intubation, invasive mechanical ventilation, or extracorporeal oxygenation were admitted to the ICU.

### 2.2. Laboratory Findings

With the exception of creatinine, all presented laboratory markers were assayed during the first five days of the patient’s hospital stay. Creatinine concentrations analyzed in this paper corresponded with specific points: hospital admission, the time of AKI diagnosis, and the maximal detected value during hospitalization and at discharge. AKI was diagnosed based on the KDIGO criteria [4] of a rise or decrease in creatinine and classified as stage 1 if the creatinine change was from 1.5 to 1.9 times baseline, stage 2 for creatinine change from 2.0 to 2.9 times baseline, and stage 3 for creatinine ≥ 3.0 times baseline. Albumin and hemoglobin concentrations were usually assayed more than once during this time period, and we decided to analyze the lowest value. The rationale for this was to obtain the most representative laboratory value for each patient during this period of time. Laboratory tests, if remeasured, showed lower values on subsequent remeasurements. This is in line with clinical judgment and practice, meaning that patients received fluid resuscitation due to dehydration, volume depletion, and shock. Next, measurements were assayed after initial fluid resuscitation. Choosing the lowest values of albumin and hemoglobin reduced the bias of including false results due to hemoconcentration.

### 2.3. Computed Tomography

To assess the severity of lung involvement, the computed tomography (CT) severity score proposed by Francone et al. was used [8]. Each lobe was evaluated independently, with a minimal rank of zero points (no involvement) to 5 points (more than 75% of parenchyma involved in the assessed lobe). The calculated sum of points for the five lobes on a CT scan of a healthy individual is 0 points, while the maximal 25 points reflects involvement of more than 75% of parenchyma in each lobe. A result of 18 or more points was found to be a strong predictor of death in the original study [8]. CT scans were performed at admission to the hospital.

### 2.4. Statistical Analysis

The investigated data overview was presented as means with standard deviations and medians with interquartile ranges (IQRs). Categorical variables are shown as the number of occurrences with percentages. Depending on the distribution condition, checked with the Shapiro–Wilk test, continuous variables were compared using a t-test or Mann–Whitney U test. The chi-squared test was performed to compare categorical data. Univariable and gradual backward multivariable logistic regression analyses were conducted to investigate significant relationships with AKI occurrence. Receiver operating characteristic (ROC) analysis was performed to compare the predictive properties of considered renal function parameters.

## 3. Results

Three hundred and twenty-three patients were included in the study (195 male and 128 female). The mean age of patients was 64.5 ± 15.3 years. The mean length of hospital stay was 12.0 ± 8.3 days. Twenty-five (7.7%) patients had CKD prior to hospitalization, whereas AKI was recognized in 111 (34.4%) patients, of which 96 participants initially had normal kidney function and 15 individuals had concomitant CKD. Stage 1 AKI was recognized in 47 (14.6%), stage 2 in 42 (13.0%), and stage 3 in 22 (6.8%) patients. AKI was diagnosed on the 5.37 ± 5.15 day of the hospital stay. Demographics, chronic conditions, CT severity, and main laboratory results of the investigated groups are presented in Table 1a,b.

In the group of 25 CKD patients, AKI criteria were fulfilled by 15 (60%) participants. Patients with superimposed AKI on CKD had a greater rise (*p* = 0.006) and decrease (*p* = 0.017) in creatinine and a significantly longer hospital stay than those with CKD but without AKI (12.5 ± 4.8 vs. 5.7 ± 3.1 days; *p* < 0.001). However, there were no more differences between the considered CKD subgroups. Moreover, AKI occurrence was not associated with higher mortality when these incidents occurred in the CKD group (OR 0.771; 95%CI: 0.136–4.391; *p* = 0.770). Thus, in further analyses, we considered CKD patients with chronically decreased kidney function as a homogenous group instead of fulfilling AKI criteria.

A comparison of the results between investigated groups with NKF, AKI, and CKD is presented in Table 2a,b.

To identify factors independently associated with AKI occurrence, univariable and then gradual backward multivariable logistic regression analyses, including significantly related variables, were performed (Table 3). In this analysis, hospitalization in the ICU was the only variable independently associated with the occurrence of AKI. After eliminating the ICU hospitalization variable in Grade 2, serum albumin and sodium concentrations were independently correlated with AKI. Lastly, in Grade 3, white blood count and serum triglyceride were associated with the occurrence of AKI.

ROC analysis was performed to investigate kidney function parameters with the highest predictive value for mortality. It showed that maximal creatinine, creatinine change during hospitalization, creatinine concentration on the day of admission, AKI stage, and the day AKI observed are valuable for predicting in-hospital death (Figure 1). Comparison analysis revealed that maximal creatinine was the best marker differentiating survivors from deceased patients, and it was significantly better than creatinine changes (*p* = 0.039), creatinine on admission (*p* < 0.001), AKI stage (*p* = 0.001), and AKI day (*p* < 0.001). Proposed cut-off point values for the abovementioned factors are presented in Table 4.

## 4. Discussion

AKI is common among critically ill patients, and this is also true for individuals with COVID-19 [9]. AKI among COVID-19 patients is of mixed origin, with dehydration and direct viral infection being the two main pathways [9,10]. Dehydration and volume depletion derive from decreased fluid intake due to sickness and elevated fluid loss secondary to perspiration and exhalation [11].

Literature concerning AKI among COVID-19 patients and its impact on prognosis reports different morbidity and mortality rates, probably due to substantial differences in healthcare organization and data reporting. Nevertheless, results consistently identify an AKI correlation with severe forms of COVID-19 and a poorer prognosis. In the North American study by Hirsch et al., AKI was defined according to KDIGO and was present in 36.6% of inpatients with COVID-19. Stages 1, 2, and 3 developed in 46.5%, 22.4%, and 31.1% of AKI patients, respectively. Of note, 14.3% of patients with stage 3 AKI required RRT, and 35% of patients with AKI died [12]. In the meta-analysis, including 15,017 inpatients with COVID-19, mostly from China and the USA, the prevalence of AKI was as high as 11.6%, with a prevalence of comorbid CKD of 9.7% and 2.58%, respectively. The use of continuous RRT (CRRT) for AKI and CKD and the overall use of CRRT were significantly more prevalent in patients with severe manifestations of COVID-19 [13]. A meta-analysis of 49,692 COVID-19 patients from several countries showed that the incidence of AKI was 10.6%, ranging from 5.4% in non-severe cases of COVID-19 to 22.1% in severe cases, with the same prevalence within the group of fatal cases [14]. This study demonstrated substantial discrepancies between analyzed subpopulations. European patients had a higher mortality compared to patients from other regions. Contrarily, in China, the incidence of AKI and the mortality rate were significantly lower compared to Europe [14]. In the meta-analysis by Kunutsor et al., the pooled incidence of AKI among COVID-19 patients was 11.0%, RRT was used in 6.8% of cases, and the prevalence of preexisting CKD was 5.2%. The incidence of AKI was higher in patients with previous CKD [15]. In a recent meta-analysis by Zhang et al. that included 153,600 patients with COVID-19, AKI was diagnosed in 18.2% of the studied population [16]. Some studies report a mortality rate among COVID-19-related AKI of 54.8% [17]. Taking the above into consideration, we decided to analyze the occurrence of AKI, its risk factors, and its impact on the prognosis of inpatients with severe pneumonia due to SARS-CoV-2 infection.

We found that individuals with AKI represent a population with extremely severe COVID-19 [12,13]. In our study group, AKI patients had a prolonged hospital stay (15.4 ± 10.7 vs. 10.7 ± 6.7 days, *p* < 0.001), more frequent ICU admissions (57.5 vs. 12.9%, *p* < 0.001), and a higher mortality rate (55.2 vs. 18.8%, *p* < 0.001) compared to patients without AKI. On univariable regression, AKI was strongly associated with a poor outcome, with a 4.794 OR for in-hospital death (95%CI: 2.906–7.906, *p* < 0.001). Those results are consistent with the fact that patients with AKI presented several other markers of critical COVID-19, e.g., broader lung parenchyma involvement measured with CT (16.4 ± 6.4 vs. 13.8 ± 5.5 points, *p* < 0.001) and higher elevation of inflammation markers—higher WBC count (14.5 ± 10.3 vs. 10.4 ± 6.9 [1 × 10^3^/mm^3^], *p* < 0.001), elevated levels of fibrinogen (552.9 ± 196.3 vs. 498.7 ± 173.3 [mg/dL], *p* = 0.043), D-Dimer (13.8 ± 26.4 vs. 6.3 ± 16.4 [mg/L], *p* < 0.001), CRP (15.3 ± 9.5 vs. 9.3 ± 8.5 [mg/dL], *p* < 0.001), and ferritin (3759.8 ± 17,055.3 vs. 1293.2 ± 2299.4 [ng/mL], *p* < 0.001)—compared to the NKF group. Markers of systemic hypoperfusion like lactate dehydrogenase (683.5 ± 517.2 vs. 501.4 ± 896.4 [U/L], *p* < 0.001) and cell destruction like creatine kinase (4111.1 ± 12,768.4 vs. 835.3 ± 2697.9 [U/L], *p* < 0.001), were higher in the AKI group compared to the NKF group. Besides AKI itself, another rationale explaining the mortality rate in the AKI group is the higher number of obese individuals (29.2 vs. 18.3%, *p* = 0.034), as obesity is one of the known major risk factors for poor prognosis among COVID-19 patients [18]. Besides well-established factors for the development of AKI among critically ill individuals, we also decided to study laboratory markers that are linked with nutritional status. Malnutrition is a known factor for morbidity and mortality among inpatients, including individuals admitted with COVID-19 [19,20]. Furthermore, malnutrition was identified as a risk factor for specific complications of COVID-19, such as persistent cognitive impairment, i.e., brain fog [21]. Contrary to our expectations, we did not find a correlation between AKI occurrence and low levels of serum lipids, which may signify malnutrition. When analyzing laboratory nutrition markers, the AKI group had higher serum cholesterol (162.2 ± 51.1 vs. 140.4 ± 45.4 mg/dL, *p* = 0.018) and triglycerides (305.2 ± 151.7 vs. 185.2 ± 83.1 mg/dL, *p* < 0.001). Elevated cholesterol in the AKI group, being a known marker for cardiovascular and renal diseases, was consistent with the trend towards a statistically significant higher frequency of coronary artery disease (18.8 vs. 11.5%, *p* = 0.085) and diabetes (30.2 vs. 20.3% *p* = 0.059) and significantly more frequent obesity in the AKI group (29.2 vs. 18.3%, *p* = 0.034) compared to the group with preserved renal function. These are known clinical risk factors for kidney function impairment [22,23]. An additional explanation for the higher level of triglycerides in the AKI group is the higher frequency of ICU admissions. Propofol was administered as a first-line sedative agent. Administration of this drug often results in elevated triglycerides due to its pharmaceutical formula [24]. Similarly to other studies where hypoalbuminemia was correlated with AKI, the AKI patients in the analyzed group had lower albumin concentrations (2.7 ± 0.5 vs. 3.0 ± 0.6 [g/dL], *p* < 0.001) than NKF patients [25]. Of note, lower albumin levels accompanied inflammation [26]. Thus, the low albumin in the AKI group may have resulted from both malnutrition and a systemic inflammatory response to SARS-CoV-2. Also, serum sodium levels were higher in the AKI group (142.2 ± 5.8 vs. 139.9 ± 3.7 [mmol/L], *p* < 0.001) than those with proper kidney function. This may partially be explained by dehydration and fluid volume depletion. In the context of COVID-19, this finding is consistent with other studies identifying hypernatremia as a risk factor for a poor prognosis in COVID-19 patients [27]. Noteworthy plasma sodium alterations are frequently observed in COVID-19. Early papers reported both hypo- and hypernatremia, whereas some individuals had both disturbances during one hospital stay [28]. The first clinical explanation for hyponatremia among COVID-19 patients was the development of the syndrome of inappropriate antidiuretic hormone secretion (SIADH) [28]. Later explanations for the cause of hyponatremia included increased serum interleukin-6 levels, centrally induced hypocortisolism, vomiting, diarrhea, intestine cell damage, and kidney involvement with proximal tubulopathy [28]. Interestingly, hypernatremia (6.8%) was observed much less frequently than hyponatremia (26.5%) but has been correlated with a longer hospital stay and a greater need for ICU and mechanical ventilation [29]. The correlation was higher when hypernatremia resulted from a prior hyponatremia event [29]. Direct kidney involvement in COVID-19 may be explained by the interaction of the SARS-CoV-2 molecules with the angiotensin-converting enzyme 2 receptor, which is highly expressed in kidney tissue [30]. Contrary to our previous research on brain fog [21], where hypophosphatemia at admission was an independent risk factor for cognitive impairment at discharge, AKI patients had higher levels of phosphates (5.5 ± 5.1 vs. 4.3 ± 2.3 mg/dL, *p* = 0.024) than the NKF group. This may be clinically explained by decreased phosphate excretion and the frequent coexistence of AKI and multiple organ failure, which is associated with various disturbances in serum phosphates, including hyperphosphatemia [31]. The poor prognosis in the AKI group of patients is supported by other studies, where hyperphosphatemia was correlated with in-hospital mortality among patients with sepsis [32,33], a prolonged ICU stay, and ICU mortality [34]. While there is evidence for hyperphosphatemia being a risk factor, and it is known that phosphates are core bone minerals and essential factors for ATP synthesis, no clear rationale has been postulated besides promoting apoptosis, inflammatory cytokine release, and enhancing oxidative stress [32]. Finally, in stepwise multivariable logistic regression, we found that serum albumin (OR = 0.115, 95%CI: 0.028–0.465, *p* = 0.002), serum sodium (OR = 1.186, 95%CI: 1.029–1.368, *p* = 0.019), and WBC (OR = 1.085, 95%CI: 1.004–1.172, *p* = 0.039) were associated with AKI occurrence. This is supported by previous findings in the literature [25,27,35].

We also found several discrepancies concerning the CKD and NKF groups. Patients with CKD had higher in-hospital mortality compared to the NKF group (64.0 vs. 18.8%, *p* < 0.001), but the CKD group had other co-existing chronic diseases like diabetes (44.0 vs. 20.3%, *p* = 0.008), coronary artery disease (48.0 vs. 11.4%, *p* < 0.001), older age (75.5 ± 12.6 vs. 64.0 ± 15.9 [y], *p* = 0.001), and a higher concentration of NT-proBNP (23,378.6 ± 32,018.2 vs. 8720.2 ± 16,937.2 [ng/dL], *p* = 0.032) than the NKF group. ICU admission was not significantly more frequent compared to the NKF group (16.0 vs. 12.9%, *p* = 0.753), as the presence of several comorbidities often resulted in ICU admission refusal due to the need to triage and avoid futile treatment. Individuals from the CKD group had a higher WBC compared to the group with NKF (15.3 ± 11.4 vs. 10.4 ± 6.9 [1 × 10^3^/mm^3^], *p* < 0.001), which may be the result of a more severe clinical course of COVID-19 in this group. A higher concentration of potassium (5.3 ± 0.9 vs. 4.7 ± 0.6 [mmol/L], *p* = 0.002) and urea (148.2 ± 94.6 vs. 53.0 ± 40.0 [mg/dL], *p* < 0.001) is the result of impairment in kidney function and potentially cell damage. Lower platelet counts in the CKD group compared to the NKF group (247.3 ± 102.5 vs. 299.3 ± 115.7 [1 × 10^3^/mm^3^], *p* = 0.036) may result from inflammation and subsequent disseminated coagulation. Higher concentrations of lactates in patients with CKD compared to the group with preserved renal function correspond with a higher mortality rate, which is reported in another study concerning risk factors in COVID-19 patients [36]. In the performed univariable logistic regression, CKD was also significantly associated with mortality (OR 4.044, 95%CI: 1.723–9.490, *p* = 0.013).

In our study, from all creatinine-related parameters, the maximal detected value of creatinine during hospital stay was the best parameter for identifying high-risk mortality patients. The proposed maximal creatinine cut-off value of 1.4 [mg/dL] corresponds to a significantly decreased eGFR < 60 mL/min/1.73 m^2^, which correlates almost to stage 2 AKI. Nevertheless, all considered dynamic AKI parameters appear to be valuable for predicting in-hospital death in severe COVID-19 infections. Our results emphasize the significant role of decreasing renal function as a marker of COVID-19 severity, which could be helpful in accurate patient stratification [37].

Despite our promising results, the work presented has some limitations. First, we used a rise or decrease in creatinine to recognize the AKI occurrence. However, only a creatinine increase was specifically proposed for the recognition of AKI in the KDIGO criteria [4]. Nevertheless, in the prospective, multicenter cohort study that enrolled 1538 participants, time-dependent kidney function recovery was used to recognize resolving or non-resolving AKI, which had a significant impact on kidney-specific long-term outcomes [38]. Thus, our approach to identifying AKI by an increase or decrease in creatinine, especially in severely ill patients with unknown nephrological history, seems fully justified [39]. Some cases of AKI are community-acquired, so the KIDIGO criteria include not only a known but also a presumed increase in serum creatinine at least 1.5 × baseline within the prior 7 days. In everyday practice, patients usually do not have their creatinine concentration results within the last week to compare values. In our study, for the diagnosis of AKI, we took into account only the change in creatinine concentration. Like other researchers, we did not take into consideration urinary output due to missing data [5,6]. Reliable monitoring of hourly diuresis is carried out in the ICU [40], but most patients in our study were hospitalized in HDU beds.

Due to the poor status of patients at admission, a medical interview was shortened, and medical charts of individuals transferred from other hospitals often lacked information. Thus, we wanted to surrogate a thorough medical history review by using the criterion of a reduced elevated creatinine concentration to increase the sensitivity of an AKI diagnosis.

Also, our study describes only the impact of AKI on in-hospital morbidity and mortality. It is well established that AKI onset among inpatients influences their status after discharge. This depends on but is not limited to the return of kidney function [38]. This is even more important for COVID-19 patients, as severe COVID-19-associated AKI was associated with worse long-term post-AKI kidney function recovery [41].

The last limitation that should be pointed out is that our research was a single-center observational study, and all examined patients were white Caucasians, resulting from the characteristics of the Polish population.

## 5. Conclusions

AKI among patients with severe COVID-19 pneumonia was correlated with a more severe lung manifestation, higher blood markers of inflammation, prolonged hospital stay, and elevated in-hospital mortality rate compared to those with NKF. CKD was also associated with a poor outcome. The maximal serum creatinine concentration turned out to be the best marker differentiating survivors from deceased patients.

## Figures and Tables

**Figure 1 jcm-13-01486-f001:**
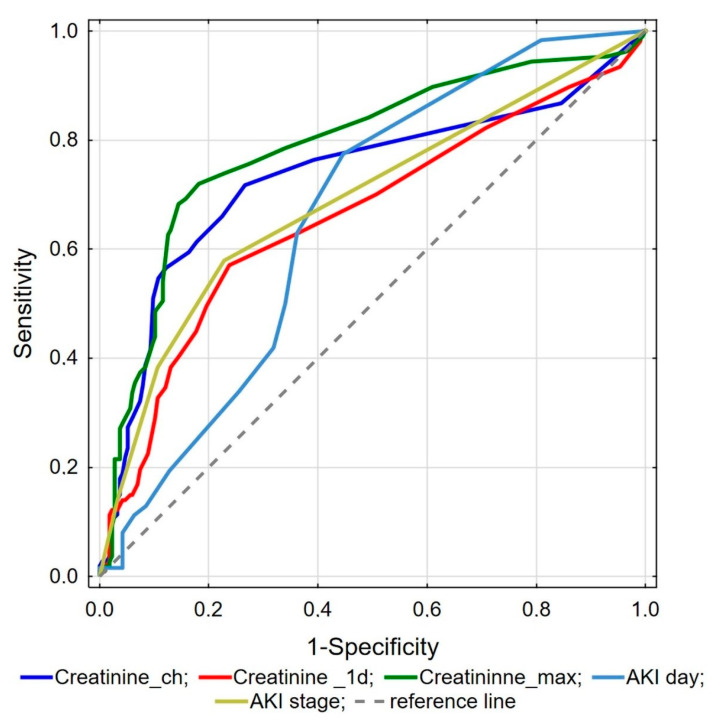
ROC analysis of kidney function parameters for mortality prediction.

**Table 1 jcm-13-01486-t001:** (**a**) Demographics, chronic conditions, CT severity, and crucial laboratory results of the investigated groups—categorical variables. Results were obtained from all patients. (**b**) Demographics, chronic conditions, CT severity, and crucial laboratory results of the investigated groups—continuous variables. Results were obtained from all patients.

(a)				
	(n)		(%)	
Hypertension	176/323		54.49	
Coronary Artery Disease	53/323		16.41	
Diabetes	81/323		25.08	
Obesity	71/323		21.98	
Chronic Obstructive Lung Disease	15/323		4.64	
Malignancy	46/323		14.24	
ICU admission	85/323		26.32	
Death	107/323		33.13	
**(b)**				
	**Mean**	**SD**	**Median**	**IQR**
Age (years)	64.47	15.33	67.00	21.00
Hospital stay (days)	12.01	8.30	10.60	9.70
Computed tomography (points)	14.52	6.08	15.00	9.00
White blood cells (1 × 10^3^/mm^3^)	12.04	8.64	9.78	7.86
Platelets (1 × 10^3^/mm^3^)	303.10	126.25	292.50	171.50
Hemoglobin (g/dL)	11.90	2.18	12.10	3.00
Sodium (mmol/L)	140.65	4.65	140.50	5.00
Potassium (mmol/L)	4.76	0.69	4.70	0.80
Calcium (mg/dL)	8.80	5.70	8.40	0.70
Phosphates (mg/dL)	4.99	3.97	4.00	2.15
Total bilirubin (mg/dL)	0.77	1.00	0.40	0.30
Alanine aminotransferase (IU/mL)	88.32	245.55	42.00	44.00
Aspartate aminotransferase (IU/mL)	126.67	441.02	49.00	46.00
Quick ratio (%)	82.16	16.12	83.00	22.00
Partial thromboplastin time (s)	44.43	21.26	38.70	14.80
Fibrinogen (mg/dL)	519.53	184.40	489.00	233.00
D-Dimer (mg/L)	9.32	21.96	1.88	5.95
Total cholesterol (mg/dL)	150.91	49.58	146.00	58.00
Triglycerides (mg/dL)	246.87	138.28	230.00	159.00
Albumin (g/dL)	2.88	0.56	2.80	0.90
C-reactive protein (mg/dL)	11.42	9.36	9.10	12.30
Procalcitonin (ng/dL)	4.24	33.06	0.23	0.81
Ferritin (ng/mL)	2318.26	10,297.63	933.50	1198.50
Lactate dehydrogenase (U/L)	595.16	883.96	418.00	338.00
Lactates (mmol/L)	2.31	2.82	2.00	1.20
NT-proBNP (ng/dL)	6678.22	14,476.01	1266.50	5218.00
Creatine kinase (U/L)	2502.53	9387.03	254.50	716.00
Urinary acid (mg/dL)	5.05	2.10	4.95	2.70
Urea (mg/dL)	73.34	57.65	55.50	58.50
Creatinine change (mg/dL)	0.68	1.01	0.30	0.80
Creatinine at admission (mg/dL)	1.24	1.04	0.90	0.60
Creatinine maximal (mg/dL)	1.62	1.34	1.00	1.10
Creatinine discharge (mg/dL)	1.25	0.98	0.90	0.60
Creatinine rise (mg/dL)	1.35	0.77	1.00	0.33
Creatinine reduction (mg/dL)	1.35	0.73	1.11	0.40

NT-proBNP—N-terminal pro-B-type natriuretic peptide.

**Table 2 jcm-13-01486-t002:** (**a**) Differences between the NKF, AKI, and CKD groups—categorical variables. (**b**) Differences between the NKF, AKI, and CKD groups—continuous variables.

(a)									
	NKF (*n* = 202)		AKI (*n* = 96)		CKD (*n* = 25)		Significance—P		
	(*n*)	(%)	(n)	(%)	(n)	(%)	(NKF:AKI)	(AKI: CKD)	(NKF:CKD)
Hypertension	104	51.5	55	57.3	17	68.0	0.348	0.331	0.118
CAD	23	11.4	18	18.8	12	48.0	0.085	0.003	<0.001
Diabetes	41	20.3	29	30.2	11	44.0	0.059	0.192	0.008
Obesity	37	18.3	28	29.2	6	24.0	0.034	0.609	0.587
COPD	9	4.5	3	3.13	3	12.0	0.757	0.102	0.133
Malignancy	30	14.8	14	14.6	2	8.0	0.951	0.520	0.544
ICU admission	26	12.9	55	57.3	(4)	16.0	<0.001	<0.001	0.753
In-hospital Death	38	18.8	53	55.2	(16)	64.0	<0.001	0.429	<0.001
**(b)**									
	**NKF** **(*n* = 202)**		**AKI** **(*n* = 96)**		**CKD** **(*n* = 25)**		**Significance—P**		
	**Mean** **±SD**	**Median** **[IQR]**	**Mean** **±SD**	**Median** **[IQR]**	**Mean** **±SD**	**Median** **[IQR]**	**(NKF: AKI)**	**(AKI: CKD)**	**(NKF: CKD)**
Age (years)	64.0 ±15.9	66.5 [23.0]	65.5 ±14.0	68.0 [17.0]	75.5 ±12.6	71.0 [18.0]	0.563	0.003	0.001
Hospital stay (days)	10.7 ±6.7	9.9 [7.7]	15.4 ±10.7	14.6 [11.7]	9.3 ±5.6	9.2 [7.2]	<0.001	0.004	0.395
Computed tomography (points)	13.8 ±5.5	15.0 [8.0]	16.4 ±6.4	18.0 [10.0]	11.5 ±7.4	9.0 [12.5]	<0.001	0.020	0.145
WBC count (1 × 10^3^/mm^3^)	10.4 ±6.9	8.8 [6.7]	14.5 ±10.3	12.4 [9.9]	15.3 ±11.4	12.4 [9.7]	<0.001	0.708	0.004
Platelets (1 × 10^3^/mm^3^)	299.3 ±115.7	296.0 [174.0]	324.9 ±147.1	299.5 [178.5]	247.3 ±102.5	245.0 [167.5]	0.494	0.041	0.036
Hemoglobin (g/dL)	12.2 ±2.1	12.3 [2.8]	11.4 ±2.3	11.5 [3.6]	11.8 ±2.0	12.0 [2.6]	0.005	0.456	0.337
Sodium (mmol/L)	139.9 ±3.7	140.0 [4.0]	142.2 ±5.8	142.0 [6.0]	141.2 ±5.3	141.0 [7.0]	<0.001	0.437	0.321
Potassium (mmol/L)	4.7 ±0.6	4.7 [0.7]	4.8 ±0.8	4.8 [0.7]	5.3 ±0.9	5.0 [1.1]	0.286	0.023	0.002
Calcium (mg/dL)	8.4 ±0.7	8.4 [0.7]	9.5 ±9.4	8.4 [0.8]	8.1 ±0.8	8.3 [0.7]	0.611	0.307	0.166
Phosphates (mg/dL)	4.3 ±2.3	3.7 [1.7]	5.5 ±5.1	4.4 [2.3]	6.3 ±3.6	5.1 [6.0]	0.024	0.506	0.172
Total bilirubin (mg/dL)	0.4 ±0.3	0.3 [0.3]	0.9 ±1.1	0.5 [0.6]	1.0 ±1.4	0.3 [1.1]	0.026	0.542	0.860
Alanine aminotransferase (IU/mL)	77.5 ±231.1	40.0 [42.0]	100.2 ±234.8	45.5 [53.0]	127.2 ±374.6	39.0 [67.0]	0.149	0.320	0.761
Aspartate aminotransferase (IU/mL)	103.3 ±373.6	40.5 [41.5]	135.7 ±316.1	59.0 [45.0]	274.0 ±1007.1	52.0 [74.0]	<0.001	0.494	0.247
Quick ratio (%)	81.4 ±17.2	83.0 [23.0]	84.5 ±13.2	84.5 [17.5]	77.2 ±18.5	76.0 [25.0]	0.196	0.042	0.383
Partial thromboplastin time (s)	42.2 ±16.8	37.9 [12.7]	48.5 ±27.9	41.2 [18.5]	43.7 ±14.0	41.4 [13.4]	0.210	0.932	0.353
Fibrinogen (mg/dL)	498.7 ±173.3	458.0 [213.0]	552.9 ±196.3	552.0 [231.0]	473.5 ±172.1	495.0 [177.5]	0.043	0.270	0.973
D-Dimer (mg/L)	6.3 ±16.4	1.5 [3.3]	13.8 ±26.4	4.1 [11.0]	15.9 ±36.1	3.8 [4.2]	<0.001	0.447	0.062
Total cholesterol (mg/dL)	140.4 ±45.4	130.0 [57.0]	162.2 ±51.1	157.5 [54.0]	123.0 ±49.7	108.0 [59.0]	0.018	0.101	0.293
Triglycerides (mg/dL)	185.2 ±83.1	169.0 [127.0]	305.2 ±151.7	276.5 [131.0]	197.4 ±157.0	132.0 [166.0]	<0.001	0.090	0.698
Albumin (g/dL)	3.0 ±0.6	3.0 [0.9]	2.7 ±0.5	2.6 [0.7]	2.7 ±0.3	2.8 [0.3]	<0.001		0.114
C-reactive protein (mg/dL)	9.3 ±8.5	6.7 [11.0]	15.3 ±9.5	14.3 [11.6]	13.0 ±10.2	10.4 [10.7]	<0.001	0.152	0.053
Procalcitonin (ng/dL)	4.6 ±43.1	0.1 [0.3]	3.8 ±11.4	0.5 [1.8]	3.6 ±8.5	0.3 [1.3]	<0.001	0.755	0.002
Ferritine (ng/mL)	1293.2 ±2299.4	825.5 [850.0]	3759.8 ±17,055.3	1553.0 [1654.0]	4267.5 ±8721.9	806.0 [3200.0]	<0.001	0.649	0.557
Lactate dehydrogenase (U/L)	501.4 ±896.4	370.0 [177.0]	683.5 ±517.2	593.0 [345.0]	924.2 ±1798.6	390.0 [517.0]	<0.001	0.096	0.770
Lactates (mmol/L)	2.9 ±5.3	1.5 [1.2]	2.0 ±0.7	2.1 [1.2]	2.7 ±0.8	2.7 [0.8]	0.087	0.093	0.032
NT-proBNP (ng/dL)	8720.2 ±16,937.2	911.0 [5422.0]	3182.9 ±5826.9	1222.5 [1881.0]	23,378.6 ±32,018.2	13,033.0 [12,342.0]	0.854	0.005	0.032
Creatine kinase (U/L)	835.3 ±2697.9	158.5 [508.5]	4111.1 ±12,768.4	496.5 [998.5]	906.0 ±1150.8	324.5 [1285.0]	0.005	0.975	0.268
Urinary acid (mg/dL)	5.2 ±1.9	5.1 [2.9]	4.8 ±2.2	4.6 [2.6]	5.7 ±2.3	5.5 [1.1]	0.324	0.419	0.621
Urea (mg/dL)	53.0 ±40.0	42.0 [28.5]	94.5 ±51.7	88.0 [58.0]	148.2 ±94.6	124.0 [111.0]	<0.001	0.002	<0.001
Creatinine change (mg/dL)	0.2 ±0.2	0.2 [0.2]	1.4 ±1.1	1.1 [1.1]	1.7 ±1.9	1.1 [2.5]	<0.001	0.696	<0.001
Creatinine at admission (mg/dL)	1.0 ±0.6	0.8 [0.3]	1.4 ±0.9	1.0 [1.0]	3.0 ±2.2	1.8 [2.7]	<0.001	< 0.001	<0.001
Creatinine maximal (mg/dL)	1.1 ±0.7	0.9 [0.3]	2.3 ±1.3	2.0 [1.5]	3.6 ±2.3	3.4 [2.8]	<0.001	0.011	<0.001
Creatinine at discharge (mg/dL)	1.0 ±0.7	0.8 [0.3]	1.5 ±1.1	1.2 [1.3]	2.4 ±1.5	1.7 [2.7]	<0.001	0.005	<0.001
Creatinine rise (mg/dL)	1.1 ±0.1	1.0 [0.2]	1.9 ±1.2	1.5 [1.3]	1.4 ±0.6	1.0 [0.5]	<0.001	0.010	0.692
Creatinine reduction (mg/dL)	1.1 ±0.1	1.0 [0.2]	1.7 ±0.8	1.6 [1.0]	1.9 ±1.8	1.2 [0.7]	<0.001	0.280	0.010

NKF—normal kidney function; AKI—acute kidney injury; CKD—chronic kidney disease; CAD—coronary artery disease; COPD—chronic obstructive pulmonary disease; WBC—white blood cell; NT-proBNP—N-terminal pro-B-type natriuretic peptide.

**Table 3 jcm-13-01486-t003:** Results of univariable and gradual backward multivariable regression analysis for AKI occurrence.

	Univariable Analysis			Multivariable Analysis			
Variable	OR	CI	*p*	Grade	OR	CI	*p*
ICU admission	7.067	4.094–12.199	<0.001	1	36.000	4.259–304.277	0.001
Hospital stay (days)	1.070	1.038–1.104	<0.001				
Severity of SARS-CoV-2 pneumonia in CT scans (points)	1.054	1.009–1.102	0.019				
Albumin (mg/dL)	0.305	0.163–0.570	<0.001	2	0.115	0.028–0.465	0.002
C-reactive protein (mg/dL)	1.074	1.044–1.105	<0.001				
Cholesterol (mg/dL)	1.010	1.001–1.019	0.039				
Hemoglobin (g/dL)	0.885	0.767–0.953	0.005				
D-Dimer (mg/L)	1.017	1.005–1.030	0.006				
Sodium (mmol/L)	1.134	1.068–1.204	<0.001	2	1.186	1.029–1.368	0.019
Triglycerides (mg/dL)	1.008	1.004–1.012	<0.001	3	1.006	1.001–1.011	0.011
White blood cells (1 × 10^3^/mm^3^)	1.049	1.018–1.080	0.002	3	1.085	1.004–1.172	0.039

**Table 4 jcm-13-01486-t004:** Results of ROC analysis in the prediction of death in the investigated population.

Variable	Cut-Off Point	Sensitivity	Specificity	AUC (95%CI)	*p*-Value
Creatinine maximal	1.4	72.0	81.9	0.788 (0.732–0.845)	<0.001
Creatinine change	0.4	71.4	73.4	0.733 (0.668–0.799)	<0.001
Creatinine at admission	1.1	57.0	76.3	0.660 (0.593–0.727	<0.001
AKI stage	1	57.9	77.2	0.690 (0.625–0.754)	<0.001
AKI day	3	77.4	55.3	0.660 (0.552–0.767)	0.004

## Data Availability

The datasets used and analyzed in this study are available from the corresponding author upon reasonable request.

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
