# Peer review of "Acute Kidney Injury and Chronic Kidney Disease and Their Impacts on Prognosis among Patients with Severe COVID-19 Pneumonia: An Expert Center Case–Cohort Study"

_jcm, 2024, doi:10.3390/jcm13051486_

Round 1
Reviewer 1 Report
Comments and Suggestions for Authors
Minor points
1. To assist clinical decision-making, the authors should show a predicted creatinine maximal “range score” specific for AKI or CDK to patient management.
2. Have the authors investigated albumin and triglycerides in ROC analysis, based on p values, seems interesting to verify.
Author Response
Authors would like to express gratitude for the time and consideration. At this moment we are not able to perform additional analyses. As we plan to publish other papers from this database, we will include Reviewers suggestion.
Minor points
- To assist clinical decision-making, the authors should show a predicted creatinine maximal “range score” specific for AKI or CDK to patient management.
Thank you for this valuable comment. We used data from anamnesis for CKD recognition without any specification of creatinine concentration because, in the presented work, AKI on CKD had no substantial impact on mortality. Moreover, considering AKI's influence on mortality, in the ROC analysis, we proposed the best cut-off value of creatinine concentration of 1.4 mg/dL (Table 4) with the optimal sensitivity and specificity (based on the Youden index) of distinguishing between survivors and the deceased patients. Thus, in our opinion, this is the best decision-making range score for AKI.
- Have the authors investigated albumin and triglycerides in ROC analysis, based on p values, seems interesting to verify.
Thank you very much for this idea. However, in this work, we focused mainly on the significance of AKI occurrence in the course of severe COVID-19. Moreover, we investigated which AKI-related parameter is superior in mortality prediction. In our study, factors associated with AKI are most likely caused/influenced by severe inflammation, hypoxemia, and administered treatment, thus secondary to the severe infection. To investigate their independent influence on AKI, further studies with non-infection AKI control group should be conducted.
Reviewer 2 Report
Comments and Suggestions for Authors
This study aims to investigate AKI occurrence and previous CKD and factors associated with clinical outcomes in hospitalized COVID-19 patients; however, there are many issues to improve this manuscript as follow.
1. Please add “retrospective” into the study design to let the readers get the study timeline.
2. Abstract: Please confirm the sentence “ICU transfer was not more frequent (16.0 vs. 25 12.9%, p=0.753)”. Since the abstract did not explain the high dependencies unit, this statement may mislead the readers.
3. Strongly recommend improving the introduction. General stories about AKI criteria development and guidelines comparison should be removed. Please specify the importance of this study which is currently vague. At the end of page2, I do not think this sentence is correct “The second goal was to identify factors associated with deterioration of kidney function among COVID patients”. What authors would like to identify factors associated with? The kidney function among COVID-19 patients? Please confirm.
4. Materials and methods:
4.1) please remove “323” from methods section.
4.2) please add the explanation in the methods why authors select the lowest value of albumin and hemoglobin levels in this study.
4.3) please check the reference and citation. Ref 8 is missing. Ref 9 in methods indicated “the original study” but it currently is KDIGO guideline.
5. Results
5.1) I am not sure whether I understand data presentation correctly. There is an incorrect number of “AKI with previous CKD”. At the end of page 3 said “15” individuals had concomitant CKD but Page 6 said “14”.
5.2) Page 6, “They had higher rise (p=0.006) and decrease (p=0.017) of creatinine”. Please clearly specify who they are. All patients diagnosed with AKI or only AKI or AKI with previous CKD patients.
5.3) The table presentation needs to be improved. Recommend combining table 2 with table 3 since table 2 presents “total number and percentages” and table 3 separately presents data according to renal function.
5.4) please confirm whether all laboratory profiles come from all patients. If not, please indicate the number of patients who obtained those data in the table too.
5.5) The most important issue is the necessary of ROC analysis in the prediction of death by using “Maximal creatinine” (that the study reports as the best marker). The authors did not mention the cut-off value of maximum numbers; therefore, in practice, we need to wait for creatinine to reach the highest value (without exact number, exact duration of admission), then the patient may recover or die. The results cannot be used in the real clinical implementation.
6. Discussion: Please confirm the statement about albumin on Page 9, line 227. Low serum albumin levels of critically ill patients are not reflected in nutritional status but related to inflammatory status, as the data presented that CRP and procalcitonin levels of patients with AKI are higher than those of NKP.
Author Response
This study aims to investigate AKI occurrence and previous CKD and factors associated with clinical outcomes in hospitalized COVID-19 patients; however, there are many issues to improve this manuscript as follow.
Authors would like to express gratitude for the time and consideration. We corrected submitted paper according to specific remarks listed below. We hope, that performed correction meet expectations of Reviewers and Editor.
- Please add “retrospective” into the study design to let the readers get the study timeline.
Thank you, we added information about retrospective nature of the study (both in abstract and methods section)
- Abstract: Please confirm the sentence “ICU transfer was not more frequent (16.0 vs. 25 12.9%, p=0.753)”. Since the abstract did not explain the high dependencies unit, this statement may mislead the readers.
Thank you, we expanded information in abstract, it is explained now that patients were transferred from HDU (high dependency unit) into ICU section more often in AKI group, while in CKD group transfer was not more common, comparing to group with normal kidney function.
- Strongly recommend improving the introduction. General stories about AKI criteria development and guidelines comparison should be removed. Please specify the importance of this study which is currently vague.
Thank you, we removed part of introduction concerning general knowledge about acute kidney injury. Now it is more straightforward.
At the end of page2, I do not think this sentence is correct “The second goal was to identify factors associated with deterioration of kidney function among COVID patients”. What authors would like to identify factors associated with? The kidney function among COVID-19 patients? Please confirm.
Thank you for this question. As you suggested, we analyzed factors associated with AKI in COVID-19 patients. In the results: “To identify factors independently associated with AKI occurrence, univariable and then gradual backward multivariable logistic regression analyses, including significantly related variables, were performed (Table 4). In this analysis, hospitalization in the ICU was the only variable independently associated with the occurrence of AKI. After eliminating the ICU hospitalization variable (grade 2), serum albumin and sodium concentrations in-dependently correlated with AKI. Lastly (grade 3), white blood count and serum triglyceride were associated with the occurrence of AKI.”
- Materials and methods:
4.1) please remove “323” from methods section.
Thank you, we deleted number of patients from this part of manuscript. Corrected sentence now is “The study included Caucasian race patients with severe COVID-19 pneumonia requiring oxygen therapy, who were hospitalized in Warsaw, Poland, during the SARS-CoV-2 variant delta pandemic outbreak from 03.2021 to 06.2021.”
4.2) please add the explanation in the methods why authors select the lowest value of albumin and hemoglobin levels in this study.
Thank you for this comment, we add sentence giving rationale for choosing lower values. Information included in “methods” is: Albumin and hemoglobin concentrations were usually assayed more than once in this period of time, and we decided to analyze the lowest value. Rationale for this was to obtain the most representative laboratory value for each patient in this period of time. Laboratory tests, if remeasured, showed lower values in subsequent remeasurements. This stay in line with clinical judgement and practice, meaning that patients received fluid resuscitation due to dehydration, volume depletion and shock. Next measurement were assayed after initial fluid resuscitation. Choosing lowest values of albumin and hemoglobin reduced the bias of including false result due to hemocontrentation.
4.3) please check the reference and citation. Ref 8 is missing. Ref 9 in methods indicated “the original study” but it currently is KDIGO guideline.
Thank you, we corrected references.
- Results
5.1) I am not sure whether I understand data presentation correctly. There is an incorrect number of “AKI with previous CKD”. At the end of page 3 said “15” individuals had concomitant CKD but Page 6 said “14”.
Thank you for this valuable comment. You are right. We have made a mistake. The corrected sentence on page 6 (actually page 5) is: “In the group of 25 CKD patients, AKI criteria were fulfilled by 15 (60%) participants.”
5.2) Page 6, “They had higher rise (p=0.006) and decrease (p=0.017) of creatinine”. Please clearly specify who they are. All patients diagnosed with AKI or only AKI or AKI with previous CKD patients.
Although the abovementioned sentence is placed in the CKD group section, according to the comment, we improved the meaning: “Patients with AKI on CKD had a higher rise (p=0.006) and decrease (p=0.017) of creatinine and significantly longer hospital stay than those with CKD but without AKI (12.5 ±4.8 vs. 5.7 ±3.1 days; p<0.001).”
5.3) The table presentation needs to be improved. Recommend combining table 2 with table 3 since table 2 presents “total number and percentages” and table 3 separately presents data according to renal function.
Thank you for this comment. In previous version of manuscript both data about all studied population (tables 2a and 2b together) and groups (3a and 3b) were presented as two tables. In draft correction most of our team suggested, that mixing categorical and continuous variables makes the table hard to understand. When resubmitting corrected manuscript we will ask Editors, how they see it and we`ll leave decision about tables to their discretion.
5.4) please confirm whether all laboratory profiles come from all patients. If not, please indicate the number of patients who obtained those data in the table too.
Thank you for this important comment. Due to extremely high volume of patients and shortcomings of medical staff we based on panels of laboratory tests. This meant, that every patients received the same tests in particular day of hospital stay as a baseline. Healthcare providers added to baseline panels more tests, if needed, according to patients` need. But the baseline was obtained all the time. We chose to include into analysis only laboratory values from baseline panels, thus mean there is no gaps in data. We decided not to analyze tests not included in panels e.g. IL-6, as this test was not a standard and dataset showed substantial gaps.
We added this information after Table 1b, showing characteristics of studied population.
5.5) The most important issue is the necessary of ROC analysis in the prediction of death by using “Maximal creatinine” (that the study reports as the best marker). The authors did not mention the cut-off value of maximum numbers; therefore, in practice, we need to wait for creatinine to reach the highest value (without exact number, exact duration of admission), then the patient may recover or die. The results cannot be used in the real clinical implementation.
Thank you for this comment. We agree with the presented point of view. However, the ROC analysis was performed, and it is graphically presented in Figure 1. Moreover, the proposed best cut-off values for considered AKI-related parameters are shown in Table 5. This is mentioned in the Discussion: “The proposed maximal creatinine cut-off value of 1.4 mg/dL corresponds to significantly decreased eGFR < 60 ml/min/1.73 m2 and almost 2-nd AKI stage”. According to the abovementioned, AKI with creatinine > 1.4 mg/dL predicts death with 72% sensitivity and almost 82% specificity (p<0.001).
- Discussion: Please confirm the statement about albumin on Page 9, line 227. Low serum albumin levels of critically ill patients are not reflected in nutritional status but related to inflammatory status, as the data presented that CRP and procalcitonin levels of patients with AKI are higher than those of NKP.
This is excellent point, thank you. We added the information, that albumin correspond with nutrition status, but also depends on inflammation, as “negative acute phase marker”, which explain better lower albumin level in NKF group, comparing to AKI group.
“Similarly to other studies where hypoalbuminemia was correlated with AKI, the AKI patients in the analyzed group had lower albumin concentration- (2.7 ±0.5 vs. 3.0 ±0.6 [g/dL], p<0.001) than NKF patients [25]. Noteworthy, lower albumin level accompanies inflammation [26]. Thus low albumin in AKI group may result from both malnutrition and systemic inflammatory response to SARS-CoV-2.”
Reviewer 3 Report
Comments and Suggestions for Authors
Dear Authors,
I would like to congratulate you on the submitted article, which is based on the analysis of a significant, well-structured, and correctly analyzed database.
The impact of COVID-19 pneumonia on renal function and the rate of complications and mortality is well documented through the analysis of an impressive database.
The discussions are pertinent and correlate with the presented results.
I congratulate you on the article.
Author Response
Dear Authors,
I would like to congratulate you on the submitted article, which is based on the analysis of a significant, well-structured, and correctly analyzed database.
The impact of COVID-19 pneumonia on renal function and the rate of complications and mortality is well documented through the analysis of an impressive database.
The discussions are pertinent and correlate with the presented results.
I congratulate you on the article.
Dear Reviewer, thank you for the time, consideration and kind words.
Reviewer 4 Report
Comments and Suggestions for Authors
First, congratulations for the work done on the impact of AKI and CKD on the prognosis of patients with severe COVID19.
However, we see many points to improve, but mainly, I find it complicated to describe a case cohort study on a topic that is currently of little interest, after the COVID19 vaccination.
The results of the study add little to the literature, and there is nothing new described in this study, patients with severe COVID-19 has a good correlation with AKI and thus higher mortality as described in other similar previous work.
Then there are other points to be clarified:
-In lines 34-35: Modify this sentence, it is not clear what is meant.
-On line 75: Define severe COVID-19.
-In lines 91 to 94: I recommend to remove these lines, the AKI criteria according to KDIGO are repeated.
-The title of the manuscript mentions the impact of AKI and CKD on the prognosis of patients with severe COVID19, but it should specify that the prognosis is only short-term, during hospital admission; it would probably be more interesting to know the long-term prognosis, at 12 or 24 months of follow-up after discharge.
I recommend making a comment on this and adding this biblical quotation
*Long-term kidney function recovery and mortality after COVID-19-associated acute kidney injury: an international multi-centre observational cohort study.
https://doi.org/10.1016/j.eclinm.2022.101724
-It would be interesting to make a comment in the discussion about the type of AKI that the patients in this cohort developed, I understand that if it is associated with hypernatremia, it is assumed that this is due to a pre-renal cause?
-Natremia alterations are a spectrum in patients with COVID-19, and there are many papers on it, expand the discussion on this topic.
*Hyponatremia and SARS-CoV-2 infection: A narrative review. Medicine (Baltimore). 2022 Aug 12; 101(32): e30061. doi: 10.1097/MD.0000000000030061
*Is There a Relationship between COVID-19 and Hyponatremia?. Medicina (Kaunas). 2021 Jan; 57(1): 55. doi: 10.3390/medicina57010055
*Sodium alterations impair the prognosis of hospitalized patients with COVID-19 pneumonia. DOI: https://doi.org/10.1530/EC-21-0411
* Sodium status and kidney involvement during COVID-19 infection. doi.org/10.1016/j.virusres.2020.198034
- I recommend a better discussion on bone mineral metabolism and phosphatemia.
As a final recommendation, I see as the main problem the low interest of the results obtained in this case-cohort study, probably improved if the long-term impact is described.
Comments on the Quality of English Language
Minor editing of English language required
Author Response
First, congratulations for the work done on the impact of AKI and CKD on the prognosis of patients with severe COVID19.
However, we see many points to improve, but mainly, I find it complicated to describe a case cohort study on a topic that is currently of little interest, after the COVID19 vaccination.
The results of the study add little to the literature, and there is nothing new described in this study, patients with severe COVID-19 has a good correlation with AKI and thus higher mortality as described in other similar previous work.
Authors would like to express gratitude for the time and consideration. We corrected submitted paper according to specific remarks listed below. We hope, that performed correction meet expectations of Reviewer and Editor.
Then there are other points to be clarified:
-In lines 34-35: Modify this sentence, it is not clear what is meant.
Thank you. As this issue was identified by more reviewers, we decided to remove broad part of introduction concerning AKI in general. Now the introduction leads clearly to the point. We hope, that this meets your expectations.
-On line 75: Define severe COVID-19.
Thank you for this important comment. Now in “methods” there is sentence explaining term “severe COVID-19” used in our study. “We enrolled into a study only patients with broad lung involvement and need for any form of oxygen therapy (simple mask, non-rebreather masks, high-flow nasal oxygen therapy, non invasive and invasive mechanical ventilation) defining “severe COVID-19 pneumonia” for the purpose of the study.”
-In lines 91 to 94: I recommend to remove these lines, the AKI criteria according to KDIGO are repeated.
Thank you. As this issue was identified by other reviewers, we decided to remove broad part of introduction concerning AKI diagnose in general. Now the introduction leads clearly to the point. We hope, that this meets your expectations.
-The title of the manuscript mentions the impact of AKI and CKD on the prognosis of patients with severe COVID19, but it should specify that the prognosis is only short-term, during hospital admission; it would probably be more interesting to know the long-term prognosis, at 12 or 24 months of follow-up after discharge. I recommend making a comment on this and adding this biblical quotation.
Thank you for raising this important matter. Unfortunately, according to Polish law we can`t reach the patients after hospital discharge without obtaining their written consent. Obtaining such consent was unpractical, as many of patients arrived to our hospital sedated, intubated, or had altered mental status secondary to the disease itself and/or profound hypoxemia. Due to high clinical significance we added paragraph in discussion/limitation concerning impact of AKI also on long-term prognosis, not only on in-hospital survival. We also added this point in “limitations” as it is important shortcoming. ” Also, our study describes only impact AKI on in-hospital morbidity and mortality. It is well established, that AKI onset among inpatients influence on their status also after discharge. This depends on, but is not limited to return of kidney function. This is even more important for COVID-19 patients, as severe COVID-19-associated AKI was associated with worse long-term post-AKI kidney function recovery”.
-It would be interesting to make a comment in the discussion about the type of AKI that the patients in this cohort developed, I understand that if it is associated with hypernatremia, it is assumed that this is due to a pre-renal cause?
This is very good point. We can`t truly answer this question and show the proportions of AKI mechanism in our study sample. Basing our opinion on literature and clinical judgement, patients involved into a study had two main causes of AKI:
- pre- renal due to volume depletion secondary to fever, low fluid intake at home, before hospital admission
- renal, due to critical illness. This would be probably majority of cases admitted to ICU. Main reason among them would be sepsis-related acute kidney injury (s-AKI).
As this is excellent remark, we added information concerning postulated pathophysiology of AKI among our patients into “Discussion” We add “AKI is common among critically ill patients, this sentence is true also for individuals with COVID-19. AKI among COVID-19 patients is of mixed origin, with dehydratation and direct viral infection being two main pathways. Dehydratation and volume depletion comes from decreased fluid intake due do sickness and elevated fluid loss secondary to perspiration and exhalation.”
Natremia alterations are a spectrum in patients with COVID-19, and there are many papers on it, expand the discussion on this topic.
Thank you for this valuable comment. Indeed, hypernatremia among COVID-19 patients can`t be explained only with dehydratation and fever. We expand discussion concerning this topic.
“It may partially be explained by dehydratation and fluid volume depletion. In the context of COVID-19 this finding is consistent with other studies identifying hypernatremia as a risk factor for poor prognosis in COVID-19 patients. Noteworthy plasma sodium alterations are frequently observed in COVID-19. Early papers reported both hypo- and hypernatremia, some of individuals had both disturbancies during one hospital stay. First clinical explanation for hyponatremia among COVID patients was syndrome of inappropriate antidiuretic hormone secretion (SIADH). Later explanation for hyponatremia were: increased serum interleukine-6 levels, centrally induced hypocortisolism, vomiting, diarrhea, intestine cell damage and kidney involvement with proximal tubulopathy. Interestingly, hyponatremia was more often (26.5%), but the less common (6.8%) hypernatremia was correlated with longer hospital stay, need for ICU and mechanical ventilation. Correlation was higher when hypernatremia resulted from prior hyponatremia. Direct kidney involvement in COVID-19 may be explained by interaction of SARS-CoV-2 molecules with the the angiotensin-converting enzyme 2 receptor, which is highly expressed in kidney tissue.”
I recommend a better discussion on bone mineral metabolism and phosphatemia.
Thank you, please find expanded discussion about impact on serum hyperphosphatemia in sepsis and COVID-19.
Contrary to our research on brain fog, where hypophosphatemia at admission was an independent risk factor for cognitive impairment at discharge, the AKI patients had higher levels of phosphates (5.5 ±5.1 vs. 4.3 ±2.3mg/dL, p=0.024) than NKF group. This may be clinically explained by decreased phosphate removal and the frequent coexistence of AKI with multiple organ failure, which is also associated with various disturbances of serum phosphates, including hyperphosphatemia. Poor prognosis of AKI group patients is supported by other studies, where hyperphosphatemia was correlated with in-hospital mortality among patients with sepsis, prolonged ICU stay and ICU mortality. While there is evidence for hyperphosphatemia being risk factors, and it is known that phosphates are core bone minerals and essential factor for ATP synthesis, no clear rationale have been postulated, beside promoting apoptosis and inflammatory cytokines enhancing oxidative stress.
As a final recommendation, I see as the main problem the low interest of the results obtained in this case-cohort study, probably improved if the long-term impact is described.
Reviewer is right, but we can`t agree at this point fully. COVID-19 is not a major health threat now, due to vaccination programs and change of SARS-CoV-2 itself into milder pathogen. But it is worth mentioning, that still COVID-19 is major threat for immunocompromised individuals. In 35- million population of Poland this would be almost 1 milion of individuals receiving immunospressive agents (due to lupus, rheumatoid arthritis, vasculitis, glomerulopathies, transplant recipients), chemotherapy for various cancers etc. Furthermore, populations in Europe show important discrepancies, when analyzing percentage of fully vaccinated citizens. Lastly, in medical ICU, where authors are working currently, around of 10 patients with COVID-19 were admitted this winter, from which two died. So, we agree that COVID-19 is not a global threat, but it`s still factor for important morbidity and mortality in internal medicine ward or ICU.
Unfortunately, as we replied above, we can`t add long term analysis of survival into this study. We add this important limitation of our work into the manuscript.
Round 2
Reviewer 4 Report
Comments and Suggestions for Authors
We would like to thank the authors for making the changes based on the requested recommendations, which have greatly improved the manuscript. For my part it could be published